# Xerotolerance: A New Property in *Exiguobacterium* Genus

**DOI:** 10.3390/microorganisms9122455

**Published:** 2021-11-28

**Authors:** María Castillo López, Beatriz Galán, Manuel Carmona, Juana María Navarro Llorens, Juli Peretó, Manuel Porcar, Luis Getino, Elías R. Olivera, José M. Luengo, Laura Castro, José Luís García

**Affiliations:** 1Microbial and Plant Biotechnology Department, Centro de Investigaciones Biológicas Margarita Salas-CSIC, Ramiro de Maeztu 9, 28040 Madrid, Spain; maria.castillo@cib.csic.es (M.C.L.); bgalan@cib.csic.es (B.G.); mcarmona@cib.csic.es (M.C.); 2Department of Biochemistry and Molecular Biology, Facultad de Ciencias Biológicas, Universidad Complutense de Madrid, Av. Complutense s/n, 28040 Madrid, Spain; joana@bio.ucm.es; 3Program for Applied Systems Biology and Synthetic Biology, Instituto de Biología Integrativa de Sistemas (I2SYSBIO) (UV-CSIC), Carrer del Catedràtic Agustín Escardino Benlloch s/n, 46980 Paterna, Spain; juli.pereto@uv.es (J.P.); manuel.porcar@uv.es (M.P.); 4Department of Biochemistry and Molecular Biology, University of Valencia, 46100 Burjassot, Spain; 5Department of Molecular Biology, Facultades de Veterinaria y Biología, Universidad de León, 24007 León, Spain; lgeta@unileon.es (L.G.); erodo@unileon.es (E.R.O.); jm.luengo@unileon.es (J.M.L.); 6Department of Applied Mathematics, Materials Science and Engineering and Electronic Technology, School of Experimental Sciences and Technology, Rey Juan Carlos University, 28933 Móstoles, Spain; laura.castro@urjc.es

**Keywords:** *Exiguobacterium*, xerotolerance, biotechnology, polyextremophile, desiccation resistance

## Abstract

The highly xerotolerant bacterium classified as *Exiguobacterium* sp. Helios isolated from a solar panel in Spain showed a close relationship to *Exiguobacterium sibiricum* 255-15 isolated from Siberian permafrost. Xerotolerance has not been previously described as a characteristic of the extremely diverse *Exiguobacterium* genus, but both strains Helios and 255-15 showed higher xerotolerance than that described in the reference xerotolerant model strain *Deinococcus* *radiodurans*. Significant changes observed in the cell morphology after their desiccation suggests that the structure of cellular surface plays an important role in xerotolerance. Apart from its remarkable resistance to desiccation, *Exiguobacterium* sp. Helios strain shows several polyextremophilic characteristics that make it a promising chassis for biotechnological applications. *Exiguobacterium* sp. Helios cells produce nanoparticles of selenium in the presence of selenite linked to its resistance mechanism. Using the *Lactobacillus* plasmid pRCR12 that harbors a cherry marker, we have developed a transformation protocol for *Exiguobacterium* sp. Helios strain, being the first time that a bacterium of *Exiguobacterium* genus has been genetically modified. The comparison of *Exiguobacterium* sp. Helios and *E. sibiricum* 255-15 genomes revealed several interesting similarities and differences. Both strains contain a complete set of competence-related DNA transformation genes, suggesting that they might have natural competence, and an incomplete set of genes involved in sporulation; moreover, these strains not produce spores, suggesting that these genes might be involved in xerotolerance.

## 1. Introduction

The genus *Exiguobacterium* belongs to the order Bacillales of the low G + C phylum Firmicutes, including a diverse group of pigmented Gram-positive facultative anaerobic bacteria with variable morphologies, from small rods to cocci. *Exiguobacterium* was proposed as a new genus almost 40 years ago [1], including 17 validly recognized species at the moment of writing this article [2,3]. Strains belonging to this genus have been isolated from a globally diverse array of environmental samples such as soil, sediments, seawater, permafrost, plant rhizosphere, glaciers, industrial effluents, and hydrothermal vents [4,5,6,7,8,9,10,11,12]. This genus is currently divided into two major groups based on taxonomic and phylogenetic analysis using 16S rRNA gene sequences: Group I comprises strains isolated from cold environments, whereas Group II includes strains from alkaline marine environments and hot springs [2].

A distinctive feature of the isolates belonging to this genus is their ability to grow under extreme environmental conditions, such as at temperatures ranging from −12 °C to 55 °C (and under nutrient-limiting situations). Some *Exiguobacterium* strains are also tolerant to other abiotic stresses, such as UV radiation, H_2_O_2_, high salt concentrations, osmotic stress, and heavy metals [13,14,15,16]. Due to these interesting properties, various isolates have been explored for biotechnological and industrial purposes, including enzyme production, agriculture applications, or bioremediation [3,17,18,19,20,21,22,23,24,25,26]. In this last sense, the genus has acquired some relevance since some strains are able to degrade 4-chloroindole and polystyrene [27,28,29].

However, none of the *Exiguobacterium* strains isolated to date has been described as xerotolerant, i.e., tolerant to desiccation. Anhydrobiosis is a phenomenon related to the partial or total desiccation of some living organisms that are capable of recovering their vital functions after rehydration. The desiccated state in prokaryotes has been widely studied, mainly due to the broad spectrum of anhydrobiosis applications [30]. In this sense, *Deinococcus radiodurans* has been considered a reference strain for xerotolerance, and it has been proposed as a biotechnological chassis and a model system to study the metabolic factors that support polyextremophilic and, particularly, irradiation and desiccation tolerance [31]. The remarkable capacity of *D. radiodurans* to survive high doses of ionizing radiation has been proposed to be an adaptation to periodic desiccation because both stress factors lead to the formation of DNA double-strand breaks [32]. Nevertheless, there are many other microorganisms that are able to colonize a wide range of extreme natural and artificial environments that have been less studied, and they might also be promising sources of bacterial chassis useful as cell factories for new biotechnological applications. In this regard, there is hardly any data on the microbial ecology of one the most currently widespread manufactured extreme structures, such as solar panels [33]. We have recently shown that solar panels in a Mediterranean city (Valencia, Spain) harbor a highly diverse microbial community with more than 500 different species per panel, most of which belong to drought-, heat-, and radiation-adapted bacterial genera [33,34,35].

In this work, we show that, using the bacterial communities of a solar panel, it is possible to isolate a large number of polyextremophilic bacteria, in particular, new xerotolerant strains. One of these new isolated bacteria was *Exiguobacterium* sp. Helios, which showed several extremophile properties, but especially high xerotolerance. Its genome sequence and several physiological characteristics are described. These findings, together with the description of the first transformation protocol for this genus, lead us to consider this new strain as a potential cell factory for biotechnological applications under conditions of extreme dryness.

## 2. Materials and Methods

### 2.1. Sample Collection

Sampling of solar panels was performed as described [33]. Briefly, harvesting of microbiota was carried out by pouring sterile phosphate-buffered saline (PBS) (pH 7.4) on the solar panel and scraping the surface with a window cleaner attached to an autoclaved silicone tube (5 mm in diameter). The resulting liquid suspension was collected by using a pipette and transferred to Falcon tubes, placed on ice, and immediately transported to the lab, where it was filtered by a hydrophilic nylon membrane (Millipore, Burlington, MA, USA) (20 µm pore size) to discard particles, most of the fungi, and inorganic debris.

### 2.2. Strains and Culture Conditions

The bacterial strains used in this work were *Escherichia coli* DH10B, *E. coli* W, *E. sibiricum* 255-15 (DSM 17290^T^), and *D. radiodurans* (DSM 20539^T^/46620^T^). *Exiguobacterium* sp. Helios and *Exiguobacterium* sp. HE 26.4 were isolated in this work in different screening of solar panels (see below). For routine cultures, *Exiguobacterium* and *E. coli* strains were grown aerobically in LB medium [36] at 30 °C or 37 °C, respectively, and *D. radiodurans* was grown in TGY medium (Tryptone (1%), glucose (0.1%), yeast extract (0.5%), pH 7.2) at 30 °C, with orbital shaking at 200 rpm. For the salinity resistance test, LB medium was supplemented with increasing concentrations of NaCl (10–150 g/L). Resistance test to polyethylene glycol (PEG) was performed in LB medium supplemented with 0.5–35% of Mw 8000 PEG. Resistance to antibiotics was tested with kanamycin (1–50 µg/mL), gentamycin (1–20 µg/mL), ampicillin (12.5–100 µg/mL), chloramphenicol (4–34 µg/mL), or spectinomycin (25–100 µg/mL), which were purchased from Sigma-Aldrich (St. Louis, MO, USA).

Minimal media M63 [37] was used for growing the cells in single carbon sources, supplemented when necessary with trace elements (nitrilotriacetic acid (1.5 mg/L); MgSO_4_·7H_2_O (3.0 mg/L); MnSO_4_·2H_2_O (0.5 mg/L); NaCl (1.0 mg/L); FeSO_4_·7H_2_O (0.1 mg/L); CoSO_4_·7H_2_O (0.18 mg/L); CaCl_2_·2H_2_O (0.1 mg/L); ZnSO_4_·7H_2_O (0.18 mg/L); CuSO_4_·7H_2_O (0.01 mg/L); KAl(SO_4_)_2_ 12H_2_O (0.02 mg/L); H_3_BO_3_ (0.01 mg/L); Na_2_MoO_4_·2H_2_O (0.01 mg/L); NiCl_2_ (0.025 mg/L); Na_2_SeO_4_ (0.3 mg/L)), vitamins (cobalamin (0.05 mg/L); pantothenic acid (0.05 mg/L); riboflavin (0.05 mg/L); pyridoxamine HCl (0.01 mg/L); biotin (0.02 mg/L); folic acid (0.02 mg/L); nicotinic acid (0.025 mg/L); *p*-aminobenzoic acid (0.05 mg/L); thiamine HCl (0.05 mg/L)), goodies (FeSO_4_ 7H_2_O (2.78 mg/L); MnCl_2_ 4H_2_O (1.98 mg/L); CoSO_4_ 7H_2_O (2.81 mg/L); CaCl_2_ 2H_2_O (1.47 mg/L); CuCl_2_ 2H_2_O (0.17 mg/L); ZnSO_4_ 7H_2_O (0.29 mg/L)), and casamino acids (0.02%). The carbons tested were arabinose (10 mM), fructose (3 mM), galactose (10 mM), glucose (0.2%), lactose (10 mM), maltose (10 mM), ribose (3 mM), sucrose (10 mM), xylose (10 mM), 3-hydroxybenzoic acid (3 mM), 3,4-dihydroxybenzoic acid (3 mM), 4-hydroxybenzoic acid (3 mM), gentisic acid (3 mM) benzoic acid (3 mM), catechol (3 mM), citric acid (0.2%), pyruvic acid (3 mM), phenylacetic acid (3 mM), and succinic acid (0.2%). All products were purchased from Merck. Growth was monitored measuring the optical density of the culture at 600 nm (OD*_600_*).

### 2.3. Xerotolerance Test for the Isolation of Xerotolerant Bacterial Strains

The isolation of xerotolerant strains was performed as follows: A number of 100 µL aliquots of microbiota samples isolated from solar panels were spread into Millipore™ membrane filters (0.45 µm pore size, 47 mm diameter, mixed cellulose esters, hydrophilic) and incubated in a stove at 37 °C with 10–15% humidity for 10 days. Filters were hydrated with 1 mL of PBS and the bacterial suspension was plated on LB agar and incubated overnight at 37 °C. The isolated colonies were cultured individually in LB and subjected to successive desiccation tests cycles in order to obtain the most xerotolerant strains. The final isolated colonies were selected and conserved at −80 °C in 20% glycerol for future uses. Validation of the xerotolerance test was performed using *E. coli* W, as a reference for low xerotolerance; and *D. radiodurans,* for high xerotolerance. To carry out the comparative tests, 100 µL of the strain cultures in rich media with an OD_600_ of 0.05 were deposited on the filters and incubated in a stove at 37 °C with 10–15% humidity for several days. The filters were hydrated with 1 mL of PBS after 3, 7, and 15 days. To quantify the survival ratio, viability cell count was performed on LB agar plates.

### 2.4. Identification of Xerotolerant Isolated Strains

The identification of the xerotolerant isolated strains was made by 16S rRNA sequencing. A 500 bp conserved fragment of 16S rRNA gene was amplified by PCR from genomic DNA, using universal primers 28F (5′-GAGTTTGATCNTGGCTCAG-3′) and 519R (5′-GTNTTACNGCGGCKGCTG-3′). PCR products were checked in 1.5% agarose gel and purified with QIAquick PCR Purification Kit. Sequencing was carried out by Secugen S.L. (Madrid, Spain). The resulting sequences were compared to the nucleotide collection at NCBI using the BLAST tool (http://blast.ncbi.nlm.nih.gov/Blast.cgi) (accessed on 23 November 2021) optimized for highly similar sequences (megablast).

### 2.5. Sequencing, Assembly, and Bioinformatic Analyses of Genome

Total DNA extraction of *Exiguobacterium sp.* Helios was performed as described [38]. The genome was sequenced by Illumina sequencer and was assembled de novo by Microbes NG (http://www.microbesng.uk) (accessed on 23 November 2021) using its standard pipeline. Briefly, the closest available reference genome was identified by using Kraken (jhu.edu), and the reads were mapped to the reference genome using BWA mem (Burrows–Wheeler Aligner, sourceforge.net) to assess the quality of the data. De novo assembly of the reads was performed using SPAdes, and the reads were mapped back to the resultant contigs, using BWA mem to obtain more quality metrics. The number of reads was 1,472,423 with a median insert size of 526 bp and mean coverage of 198.5. The number of contigs delivered was 46 with an N50 of 64,238, and the largest contig size obtained was 1,262,877 bp. A total of 1,466,367 reads were mapped back to the final assembly, being 99.58% of the total number of reads. To achieve higher genome assembly, the resulting contigs were aligned with the genome of the *E. sibiricum* 255-15 using the default settings from Mauve with Geneious v 2020.0 software (http://www.geneious.com) (accessed on 23 November 2021). To enhance genome quality, we further sequenced the genome by Nanopore. A genomic library was created with the 1D Native Barcoding genomic DNA Barcode kit and run through the flow cell FLO-MIN-106D v R9 in a MinION equipment. The number of reads obtained was 105,656 with a median insert size of 5,722 and average quality of 11.47. The assembly was performed using the Galaxy Community Hub (https://galaxyproject.org/) (accessed on 23 November 2021), first selecting the reads longer than 1 kb and with a quality bigger than 10 using Filtlong software (v 0.2.0) (https://github.com/rrwick/Filtlong) (accessed on 16 November 2021) and comparing them with the Illumina reads using Unicycler (v 0.4.8) (https://github.com/rrwick/Unicycler/releases/tag/v0.4.8) (accessed on 16 November 2021) with the standard parameters. This resulted in two contigs of 3,042,803 and 110,506 bp. The genome was structurally annotated using the RAST Server [39], and automated genome annotation system, functions, names, and general properties of gene products were predicted using this method. Average nucleotide identity (ANI) was calculated using reciprocal best hits (two-way ANI) between two genomic datasets in an online tool developed at Kostas lab [40]. Phylogenetic analyses were performed using Geneious v *2020.0* software (http://www.geneious.com) (accessed on 23 November 2021). The circular map of plasmid pMCEX was created using SnapGene software (GSL Biotech LLC, San Diego, CA, USA). Comparative analyses were carried out using BLAST software at NCBI (https://blast.ncbi.nlm.nih.gov/Blast.cgi) (accessed on 23 November 2021). The genome project has been deposited at GenBank under the accession numbers CP053557 (*Exiguobacterium* sp. Helios chromosome) and CP053558.1 (*Exiguobacterium* sp. Helios plasmid pMCEX).

### 2.6. Genetic Manipulations

The shuttle vector pRCR12 (Solmeglas S. L., Madrid, Spain) was transformed successfully into competent cells of *Exiguobacterium* sp. Helios. pRCR12 contains the replicon of plasmid pSH71 from *Lactococcus lactis* and carries the *mrfp* gene encoding the mCherry protein under the control of the *P_x_* promoter of *Streptococcus pneumoniae* and a *cat* gene conferring chloramphenicol resistance [41]. The transformation was carried out as follows: Stationary-phase cultures were collected and washed with sterile cold water three times to prepare the electrocompetent cells. Electroporation was carried out in a Bio-Rad Gene Pulser/Pulse Controller (200 Ω, 25 μF and 1.5 kV/cm) using 1 μg of plasmid pRCR12, extracted from *E. coli* DH5α. Cells were recovered after 2 h in LB at 30 °C with agitation at 200 rpm and selected on LB plates supplemented with chloramphenicol at 20 µg/mL for *Exiguobacterium* sp. Helios strain and at 5 µg/mL for *E. coli* DH10B.

### 2.7. Microscopy

Cultures of *Exiguobacterium* sp. Helios were harvested at early exponential phase (OD_600_ = 1.5) and stationary phase (OD_600_ = 6.0); preparations of these cultures were photographed with a Leica DFC345 FX optical microscopy and processed by the LAS V4.2 software. We used ImageJ software [42] to measure the cell size.

To prepare the cells for transmission electron microscopy (TEM), the cultures of *Exiguobacterium* sp. Helios were harvested, washed twice in PBS, and fixed in 3% (*w*/*v*) glutaraldehyde in PBS for 1 h. Afterwards, cells were washed again several times in PBS and then suspended in 1% (*w*/*v*) OsO_4_ and 0.8% (*w*/*v*) C_6_N_6_FeK_3_ for 1 h at 4 °C. After that, cells were washed in PBS until the supernatant was clear and then gradually dehydrated in ethanol (30%, 50%, 70%, 90%, and 100% (*v*/*v*); 30 min each) and finally embedded in Spurr ERL-4221 resin, first in a 1:1 ethanol: resin suspension for 30 min and then twice in resin alone, 30 min each time. Finally, the cells embedded in the resin were transferred to clean and dry beads and polymerized at 60 °C overnight. Ultrathin sections (thickness 70 nm) were cut with a microtome using a Diatome diamond knife. The sections were picked up with 400 mesh cupper grids coated with a layer of carbon and subsequently observed in a JEOL JEM-1230 electron microscope (Jeol Ltd., Akishima, Japan).

### 2.8. Metal Resistance and Formation of SeNPs

To study the metal and metalloid resistance of strains, we used Tris-Minimal Medium (6.06 g/L Tris-HCl; 4.68 g/L NaCl; 1.49 g/L KCl; 1.07 g/L NH_4_Cl; 0.43 g/L Na_2_SO_4_; 0.2 g/L MgCl_2_ 6H_2_O; 0.03 g/L Ca_2_Cl 2H_2_O; 0.23 g/L Na_2_HPO_4_ 12H_2_O; 0.005 g/L Fe(III)NH_4_ citrate; 1 µL/L 25% HCl; 70 µg/mL ZnCl_2_; 100 µg/mL MnCl_2_ 4H_2_O; 60 µg/mL H_3_BO_3_; 200 µg/mL CoCl_2_ 6H_2_O; 20 µg/mL CuCl_2_ 2H_2_O; 20 µg/mL NiCl_2_ 6H_2_O; 40 µg/mL Na_2_MoO_4_ 2H_2_O) [43] supplemented with 1 g/L of yeast extract and the appropriate metal (Sigma-Aldrich). The concentrations of metals used were 0.62–5.0 mM NiCl_2_, 0.62–5.0 mM ZnCl_2_, 0.62–5.0 mM K_2_TeO_3_, 0.62–5.0 mM NaAsO_2_, 5–50 mM Na_3_AsO_4_, 0.1–1.2 mM CuSO_4_, 0.62–5.0 mM CdCl_2_, 0.62–5.0 mM AgNO_3_, 0.62–10.0 mM Pb(NO_3_)_2_, or 0.05–50.0 mM K_2_SeO_3_.

To establish the selenite tolerance of *Exiguobacterium* sp. Helios strain, cells were grown in 10 mL of LB in 50 mL Falcon at 30 °C with orbital shaking at 200 rpm using 0–50 mM Na_2_SeO_3_. After 48 h of incubation, selenite resistance was monitored and cell growth was determined. In addition, we also tested the ability of *Exiguobacterium* sp. Helios to reduce selenite to elemental selenium Se (0) monitored by red color formation. *Exiguobacterium* sp. Helios was grown in LB medium supplemented with 1 mM selenite for 24 h at 30 °C with orbital shaking at 200 rpm, with the appropriate controls.

### 2.9. Characterization of Selenium Nanoparticles

For transmission electron microscopy (TEM) observation, the samples were prepared by placing drops of the *Exiguobacterium* sp. Helios cell cultures onto carbon-coated copper grids and allowing the solvent to evaporate. TEM observations were performed on a JEOL model JEM-2100 instrument operated at an accelerating voltage of 200 kV. The chemical composition of the SeNPs observed was determined by energy-dispersive X-ray spectroscopy (EDX) as previously described [44]. The size of the SeNPs was determined by using the ImageJ software [42].

## 3. Results

### 3.1. Analysis of Exiguobacterium sp. Helios Genome

*Exiguobacterium* sp. Helios strain genome was sequenced by Illumina and assembled in a two-scaffolds chromosome of 3,042,310 bp with a G + C content of 47.1% and one plasmid named pMCEX of 110,560 bp with a G + C content of 41.6% (Figure 1).

The whole *Exiguobacterium* sp. Helios genome contains 3241 open reading frames (ORFs), of which 2144 (66%) can be assigned to putative functions, 69 tRNAs, and 26 rRNA operons.

Plasmid pMCEX contains 95 ORFs, but remarkably, only 37 ORFs display homology at the NCBI protein database (Appendix A). Among them, *HNY42_RS1605* contains an LD-carboxypeptidase domain encoding a putative muramoyl tetrapeptide carboxypeptidase involved in recycling of peptidoglycan (PG). Gene *HNY42_RS16150* codes a putative ParE toxin of type II toxin-antitoxin system RelE/ParE consisting of a stable toxin and a small, labile antitoxin. Under unfavorable conditions, the antitoxin is degraded, leading to activation of the toxin and resulting in growth arrest, possibly also in bacterial programmed cell death. Gene *HNY42_RS15810* codes a ComEC-like protein that could be involved in late natural competence.

*Exiguobacterium* sp. Helios genome has two large regions involved in the synthesis of flagella (Appendix A) that are organized in two clusters: *HNY42_RS11055-HNY42_RS11200* and *HNY42_RS13935-HNY42_RS14085* covering a 23.9 kb and 29.4 kb region, respectively. The presence of these genes suggests that this strain is motile as *E. sibiricum*.

Using as reference the genes of *Bacillus subtilis* involved in natural competence, we detected the existence of a complete DNA-uptake machinery in *Exiguobacterium* sp. Helios, suggesting that this strain has the ability to acquire exogenous DNA from the medium. These predicted competence genes are not clustered together and are scattered throughout the chromosome (Table 1). The *HNY42_RS05970-HNY42_RS05985* region encodes homologous proteins to ComGC known to form pseudopilins responsible for binding the exogenous dsDNA and some additional proteins required for DNA internalization in Firmicutes, such as ComGA and ComGB. *HNY42_RS05435 and HNY42_RS05440* encode ComEA and ComEC homologous proteins. In Gram-positive microorganisms, ComEA delivers the dsDNA to a protein that generates ssDNA, presumably an AddAB, before its internalization through a transmembrane pore formed by ComEC. *HNY42_RS05235* and *HNY42_RS05240* genes encode the putative AddAB homologous proteins. Internalization of ssDNA is presumably driven by the ATP-dependent translocase ComFA (coded by *HNY42_RS14095*). The RecA homologous protein that polymerizes on ssDNA and promotes a homology search along the chromosome is encoded by *HNY42_RS06690*. Moreover, there are two most probably paralogous genes, *HNY42_RS05215* and *HNY42_RS13920*, encoding the ComK regulator responsible of activating the expression of genes involved in natural competence in *B. subtilis*.

On the other hand, it has been described that a conserved genomic signature of about 50 genes coding the minimal machinery for endosporulation enables to distinguish endospore-forming organisms [45]. *Exiguobacterium* sp. Helios strain contains only 17 genes of this genomic signature, i.e., this strain only has 34% of the genes that are currently required to sporulate, suggesting that it is not an endospore-forming bacterium, but this observation raises a question about the possible association of these genes to xerotolerance (Table 2).

The large number of genes related to carbohydrate metabolism annotated in the chromosome of *Exiguobacterium* sp. Helios suggest a remarkable ability to use several sugars and polymeric carbohydrates as carbon sources. For instance, among them, *HNY42_RS05125* and *HNY42_RS00620* genes encode a putative sucrose-6-phosphate hydrolase and a putative fructokinase, respectively, most probably involved in sucrose catabolism. Table 1 shows the sugars that can be used as carbon and energy sources by this strain.

The genome analysis predicted that *Exiguobacterium* sp. Helios contains an 11,679 kb cluster (*HNY42_RS03440*-*HNY42_RS03510*) encoding the complete degradation pathway of phenylacetic acid. *HNY42_00705*, *HNY42_07190, HNY42_08240, HNY42_08800,* and *HNY42_14205* genes annotated as VOC family proteins and/or class I extradiol dioxygenases could be involved in the degradation of other aromatic compounds, such as 3,4-hydroxybenzoate, 4-hydroxybenzoate and benzoate (Table 3).

Regarding stress-resistance genes, *Exiguobacterium* sp. Helios genome contains *HNY42_RS02725,* coding a putative transport system of glycine betaine (N,N,N-trimethylglycine) which is a very efficient osmolyte that accumulates at high cytoplasmic concentrations in response to osmotic stress [46]. *HNY42_RS02725* encodes a homologous protein to OpuD from *B. subtilis* (50% identity in amino acid sequence) that is a BCCT type carrier that uptakes glycine betaine from the environment. Locus *HNY42_RS05230* encodes a putative bacterial globine. Globin-coupled sensors (GCSs) were first described as regulators of the aerotactic responses in *B. subtilis* and *Halobacterium salinarum* [47]. Some of the main roles assigned to bacterial globins are in oxygen and nitric oxide (NO) metabolism similar to the protection from the stresses caused by exposure to nitric oxide [48].

The sigma B factor coded by the *HNY42_RS00075* gene and its accessory regulatory proteins anti-sigma factors RsbW and RsbV, coded by the *HNY42_RS00080* and *HNY42_RS00085* genes, respectively, are present in *Exiguobacterium* sp. Helios genome. The sigma B factor is known to control general stress regulons in different bacteria that are activated by the exposure to either physical or nutritional stress.

*Exiguobacterium* sp. Helios genome encodes four specific DinB homologous (*HNY42_RS08860*, *HNY42_RS08570*, *HNY42_RS01380,* and *HNY42_RS13495* genes). The DNA damage-inducible (din) protein family is overrepresented in *D. radiodurans*, with 13 homologous genes, whose expression is highly induced in response to gamma radiation (γ-radiation) and mitomycin C (MMC) exposure [49].

The *HNY42_RS04520* gene encodes a PspA/IM30 family protein whose members appear to have membrane protective activity by membrane binding to preserve membrane integrity [50].

There are five genes encoding small (66 amino acids) putative cold shock proteins (Csp) (*HNY42_RS10205*-*HNY42_RS10210*-*HNY42_RS10215*; *HNY42_RS13925* and *HNY42_00710* genes). Many bacteria produce these small proteins as a response to rapid temperature downshift (cold shock). However, some Csps are non-cold-inducible and they are reported to be involved in cellular processes to promote normal growth and stress adaptation responses. Csps have been shown to contribute to osmotic, oxidative, starvation, pH, and ethanol stress tolerance, as well as to host cell invasion [51].

Concerning the genes that confer xerotolerance in other organisms, we have not found in *Exiguobacterium* sp. Helios similar genes to those described in the five pathways described for the synthesis of trehalose [52], nor those described for the synthesis of betaine [53] and sucrose [54]. Therefore, we assume that this strain does not produce trehalose, betaine, or sucrose to become xerotolerant. However, *Exiguobacterium* sp. Helios has the genes responsible for the synthesis of proline, and thus, we cannot discard the production of proline as a xerotolerance mechanism. On the other hand, *Exiguobacterium* sp. Helios does not have genes encoding proteins similar to the late embryogenesis abundant (LEA) proteins that protect some *Azotobacter vinelandii* [55] or *D. radiodurans* [56] against drought and other stresses. Nevertheless, Helios contains a gene HNY42_14175 that encodes a hypothetical protein containing a small LEA_2 region.

### 3.2. Genome Comparison of Exiguobacterium sp. Helios and E. sibiricum 255-15

As mentioned above, the closest genome sequence of *Exiguobacterium* sp. Helios strain is that from *E. sibiricum* 255-15, which was isolated from the Siberian permafrost [4]. *E. sibiricum* 255-15 genome consists of 3,040,786 bp (accession number CPO01022) and two plasmids: pEXIG01 of 4.8 kb (accession number CPO01023) and pEXIG02 of 1.7 kb (accession number CPO01024) encoding five and two unknown proteins, respectively. The annotation in RAST server reveals 3102 coding regions, 904 of which are hypothetical proteins. The typical Mauve alignment (Appendix A) revealed that *Exiguobacterium* sp. Helios shares similar genomic arrangements with *E. sibiricum* but has some significant differences that are shown in Appendix A. Several regions within the *Exiguobacterium* sp. Helios strain genome are not found in *E. sibiricum,* according to Mauve alignment and the basic local alignment search tool (Blast) at NCBI. One example is the existence of the biosynthetic cluster *HNY42_RS07465-HNY42_RS07515* that contains the genes responsible for the synthesis of the iron siderophore petrobactin. The two genes forming a putative operon, *HNY42_RS07660-HNY42_RS07700*, encode spore coat proteins that could be involved in the desiccation tolerance of the strain. *HNY42_RS08215* gene encodes a cupin, also called seed storage protein, that usually serves as biological reserve of amino acids and metals. The conserved domain, comprising a six-stranded beta barrel structure, is typically found in the phyla Firmicutes. The cupin superfamily is a group of functionally diverse proteins that are found in all three domains of life: Archaea, Eubacteria, and Eukaryota. Regarding carbohydrate catabolism, *Exiguobacterium* sp. Helios contains 38 glycoside hydrolases, 22 glycosyl transferases, 1 polysaccharide lyase, 3 carbohydrate esterase, and 12 carbohydrate-binding module family proteins, according to CAZy Database. The number of enzymes that degrade, modify, or create glycosidic bonds found in *E. sibiricum* is slightly lower, being 26 glycoside hydrolases, 20 glycosyl transferases, 1 polysaccharide lyase, 3 carbohydrate esterase, and 11 carbohydrate-binding module family protein.

Curiously, another interesting genomic difference is that *Exiguobacterium* sp. Helios strain does not have a CRISPR-CAS system, as is *E. sibiricum* coded by the *EXIG_RS00985*-*EXIG_RS1015* genes (Appendix A). The CRISPR-CAS system is also present in other related strains as *E. antarticum*.

Finally, *E. sibiricum* genome contains 27 putative transposases that are not present in *Exiguobacterium* sp. Helios strain. A lower number of transposases is beneficial in order to use it as a biotechnological robust chassis, since transposases pose a threat to the organisms as the sequence specificity of transposase insertions is usually low, and transpositions may disrupt vital genes.

### 3.3. Isolation, Identification, Growth Conditions, and Phenotypical Characterization of Exiguobacterium sp. Helios

One of the xerotolerant strains isolated from the solar panels was identified as *Exiguobacterium* sp. according to its 16S rRNA. This strain named Helios was selected for this study since no representative of this genus has been described as xerotolerant bacterium to date. Present work was performed with the strain *Exiguobacterium* sp. Helios due to its higher resistance to desiccation.

We generated a maximum-likelihood phylogeny of 16S rRNA gene sequences from eighteen representative strains of *Exiguobacterium* spp., for which complete genomes are available at the NCBI, and one typed strain of *Bacillus indicus* as outgroup. *Exiguobacterium* sp. Helios strain was positioned in Clade I, along with *Exiguobacterium* sp. 9AN and *E. sibiricum* 255-15 (Figure 2).

*E. sibiricum* 255-15 and *Exiguobacterium* sp. Helios strain were analyzed for average nucleotide identity (ANI) and average amino acid identity (AAI) to provide interspecies relationship. The *E. sibiricum* 255-15 strain was the closest strain which had a complete genome uploaded to the NCBI database. The comparison renders 89.75% average nucleotide identity (ANI) and 94.75% average amino acid identity (AAI). Based on these data, these strains do not seem to belong to the same species [40].

*Exiguobacterium* sp. Helios is a Gram-positive bacillary rod-shaped and nonsporulating facultative anaerobe. Its optimal growth temperature is 30 °C, although it can grow efficiently between 4 °C and 42 °C. When observed by optic microscopy, *Exiguobacterium* sp. Helios strain showed different phenotypes depending on the growth phase. Thus, cells at exponential phase have a bacillary/rod shape (Figure 3a), while they become coccoid at stationary phase (Figure 3b). Moreover, a difference in the length of the cells can be observed at exponential phase (average of 3.17 μm) and at stationary phase (average of 1.41 μm) (Figure 3c).

This strain is able to grow in rich and minimal media using different substrates as sole carbon and energy sources (Table 3). *Exiguobacterium* sp. Helios cells cannot grow at concentrations of NaCl higher than 60 g/L, such as other *Exiguobacterium* strains from Clade I that are only moderately resistant to salt stress. However, it can grow up to an OD*_600_* of 2.0 on LB medium containing 35% PEG 8000, i.e., under a low water activity of 0.982 [57].

In order to investigate the susceptibility to antibiotics to carry out genetic modifications of *Exiguobacterium* sp. Helios, we determined the minimum inhibitory concentration (MIC) for some commonly used antibiotics (Table 4).

Taking into account the antibiotic resistances of *Exiguobacterium* sp. Helios, we selected the plasmid pRCR12 carrying a *cat* gene conferring chloramphenicol resistance to develop a method to transform this strain. This expression vector contains the replicon of plasmid pSH71 from *L. lactis* and carries the *mrfp* gene encoding the mCherry protein under the control of the *P_x_* promoter of *S. pneumoniae* that allows a rapid detection of the plasmid. Using this plasmid, we obtained recombinant *Exiguobacterium* sp. Helios (pRCR12) colonies showing a red phenotype, indicating that the *mrfp* gene was expressed under the *P_x_* promoter in this bacterium (Appendix A).

### 3.4. Metal Resistance of Exiguobacterium sp. Helios

Bacteria that resist extreme conditions often present other capacities, for instance, the ability to resist high concentrations of some toxic metals and metalloids. In this sense, we tested the capacity of *Exiguobacterium* sp. Helios strain to grow in the presence of these compounds. Table 5 shows that the strain has moderate resistance to some metals and metalloids, especially selenite.

The resistance of *Exiguobacterium* sp. Helios to selenite was analyzed in detail. When *Exiguobacterium* sp. Helios was grown in LB containing 1 mM selenite at 37 °C, the culture acquired a red color, suggesting the reduction of selenite to elemental selenium (Appendix A). No color was observed in the absence of bacterial cells, suggesting a role of this strain in selenite reduction. In fact, *Exiguobacterium* sp. Helios was able to grow in the presence of selenite up to 20 mM at 37 °C, indicating that the resistance is close to that reported for highly tolerant selenite strains such as *Comamonas testosteroni* S44 [58], *Pseudomonas moraviensis* [59], or *Vibrio natriegens* [60].

In addition, we also tested the ability of this strain to reduce selenite to elemental Se (0) as a proof of concept for developing a useful biotechnological process producing Se nanoparticles (SeNPs). We observed electron-dense nanospheres in cells of *Exiguobacterium* sp. Helios after 24 h of growth at 30 °C in LB containing 1 mM selenite (Figure 4a,b). The EDX analysis showed that the nanoparticles presented the specific Se peak (Figure 4c). The selected area electron diffraction (SAED) pattern of the nanoparticles showed a diffuse halo, indicating that selenium is present in its amorphous form (Figure 4c, inset). SeNPs appeared as spherical nanoparticles with an average size of 162 ± 57 nm (Figure 4d).

### 3.5. Xerotolerance of Exiguobacterium sp. Helios

A desiccation assay was performed with *Exiguobacterium* sp. Helios, *E. sibiricum*, *E. coli*, and *D. radiodurans.* In addition, we tested the performance of another *Exiguobacterium* sp. HE 26.4 strain isolated from the same solar panel, but that was classified within Clade II. In these assays, we used the cells obtained from cultures at early exponential and stationary phases to see the influence of the growth phase (Figure 5a,b). The results revealed that the xerotolerance of *Exiguobacterium* sp. Helios strain was very high, and similar to that exhibited by the model xerotolerant strain *D. radiodurans,* which, in our conditions, showed a survival score of around 80% after 15 days (data not shown). Interestingly, although *E. sibiricum* and *Exiguobcaterium* sp. HE 26.4 showed a high xerotolerance, it was lower than that of *Exiguobacterium* sp. Helios strain. As expected for the non-xerotolerant control, after 3 days, no survival of *E. coli* cells was detected (data not shown). Figure 5b shows that xerotolerance is higher at stationary phase on the *Exiguobacterium* strains.

Samples of *Exiguobacterium* sp. Helios obtained from different growth phases and before or after desiccation were observed by TEM (Figure 6). The images showed a large number of coccoid cells at the stationary phase, and remarkably, the cell envelope of these coccoid cells was thicker than that of bacillary-shaped cells. Taking into account that we have observed a higher xerotolerance when cells enter the stationary phase, the acquisition of a coccoid thickened-cell envelope phenotype might be required to improve their xerotolerance.

## 4. Discussion

The genus *Exiguobacterium* was first described by Collins et al. [1], who characterized the species *E. aurantiacum*. Since then, many other species have been described, but a fundamental issue that contributes to the interest for this genus was the discovery of *E. sibiricum* from a 43.6 m deep geological layer in the permafrost core of Kolyma Lowland (northeast of Siberia) that is dated 2–3 million years old, i.e., deposited from late Pliocene and Pleistocene periods [2,4]. Earth’s permafrost is characterized by low carbon availability, low water availability, and exposure to gamma radiation produced by soil minerals. Therefore, only special microorganisms are able to survive in these conditions [61]. Whereas extremophiles are usually defined by one extreme, many natural environments, for instance permafrost, have several extreme conditions that act as selection pressures, and thus, an increasing number of species and strains isolated from these extreme environments have been found to tolerate multiple extremes [62]. Such organisms are known as polyextremophiles, and one of the best-known examples is *D. radiodurans*, recognized for its ability to withstand a multitude of extreme environments including severe cold, dehydration, vacuum, acid, and intense radiation [62]. The removal of water through heat and air-drying damages membranes, proteins, and nucleic acids and should be lethal to the majority of organisms. Nevertheless, in addition to the resistance to desiccation exhibited by spores and akinetes developed by some bacteria and cyanobacteria, respectively, it is interesting to know that also some vegetative cells can survive extreme desiccation without forming these structures [63].

Although many natural complex extreme environments can be found on the Earth, we have tried to investigate if the microorganisms can be adapted to manufactured structures supporting extreme environmental conditions, such as solar panels [33]. Surprisingly, solar panels harbor a highly diverse microbial community, which can provide potential polyextremophiles [34,35]. For many hours a day, solar panels are subjected to absolute dryness and irradiation and, therefore, one might expect that highly desiccation-tolerant bacteria can be found there. 

A simple screening procedure allowed us to isolate several bacteria from solar panels that were able to survive after several cycles of extreme desiccation. The procedure was validated using *D. radiodurans* as a positive control and *E. coli* as a negative one. Although the global microbiome analysis of the solar panels did not reveal the presence of a significant number of bacteria of *Exiguobacterium* genus [33], one of the most xerotolerant isolated bacteria from the solar panels was classified by 16S rRNA analysis as *Exiguobacterium* sp. Helios, phylogenetically very close to *E. sibiricum*. It is worth mentioning that additional screening tests of solar panels have allowed us to isolate another *Exiguobacterium* strain named HE 26.4 as well as many other xerotolerant strains that still remain to be fully characterized (Appendix A).

Considering that none of the *Exiguobacterium* species have been described as xerotolerant so far, we investigated whether this property was specific to *Exiguobacterium* strains isolated from solar panels or if it could be shared by *Exiguobacterium* sp. Helios closest relative, i.e., *E. sibiricum.* Figure 4 shows that, while the xerotolerance of *Exiguobacterium* sp. Helios is higher than that of *E. sibiricum*, this emblematic strain can be also considered as xerotolerant, adding a new extremophile property to *E. sibiricum*. Although it has been shown that *E. sibiricum* can survive under low water activity conditions, this skill does not necessarily confer it the possibility to survive extreme desiccation conditions. In fact, *Exiguobacterium* sp. Helios can grow on a medium containing 35% PEG 8000 (i.e., under matric water stress), but can only grow very poorly under salt concentrations higher than 60 g/L (i.e., under osmotic water stress). This means than the adaptation to low water activity conditions depends on the type of osmotic perturbations (i.e., electrolyte or nonelectrolyte osmotic stress (matrix stress) caused to the cells), since the adaptation mechanisms are different. On the other hand, ice formation increases solute concentration by decreasing the amount of free water available, resulting in an environment with low water activity, similar to desiccated and salt-stressed environments. The low water activity of 0.90 encountered in permafrost corresponds to an increase in solute concentration to 2.79 M NaCl (5 osm) [64]. However, in these conditions neither *Exiguobacterium* sp. Helios strain nor *E. sibiricum* [4] can grow.

Low water activity requires an increased amount of turgor pressure to maintain cellular respiration and provide the required energy for cellular processes to take place. In this sense, it has been observed that *E. sibiricum* has a thicker cell envelope, allowing withstanding high turgor pressures [65]. This appears to be also the case for *Exiguobacterium* sp. Helios, since the cells surviving desiccation have a coccoid morphology with a thick cellular envelope. We hypothesize that *Exiguobacterium* sp. Helios is able to generate this morphology during its growth, particularly during the stationary phase, and only the cells having a thicker cellular envelope will stand the desiccation conditions. This hypothesis agrees with the observation that *Exiguobacterium* sp. Helios cells recovered by rehydration show a coccoid phenotype that soon changes to a bacillary phenotype when the cells start growing again. The development of thick, multilayered envelopes rich in polysaccharides, lipids, and proteins was described in the field- and laboratory-desiccated cells of the cyanobacteria resistant to desiccation, *Chroococcidiopsis* spp. [66]. The genes involved in this phenotype remain unknown, but it was interesting to notice that *Exiguobacterium* sp. Helios and *E. sibiricum* strains contain a number of genes related to sporulation, in spite of the fact that *Exiguobacterium* bacteria do not produce spores. In addition, the analysis of the *Exiguobacterium* sp. Helios genome allowed the identification of several genes that could be involved in stress resistance. Whether all these genes are responsible for the formation of the thick envelope phenotype and the xerotolerance is a matter of further research to identify the genetic basis of anhydrobiotic adaptation.

## Figures and Tables

**Figure 1 microorganisms-09-02455-f001:**
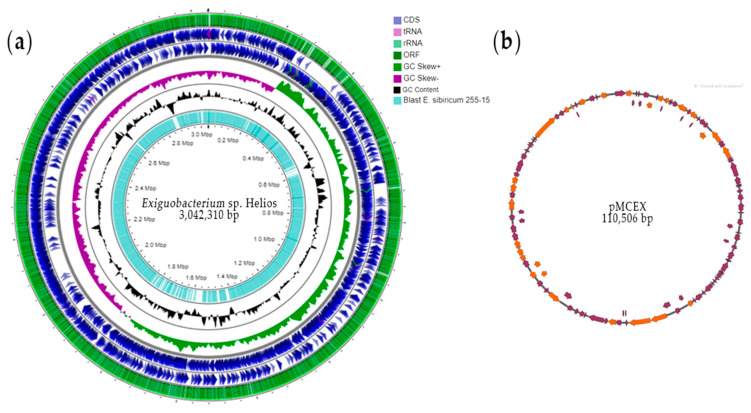
Circular map of the genome of *Exiguobacterium* sp. Helios and its plasmid pMCEX. (**a**) From outer to inner circles: The first four circles show the coding sequence (CDS), transfer ribonucleic acid (tRNA), ribosomal ribonucleic acid (rRNA), and open reading frames (ORF). The fifth circle demonstrates the GC skew curve (positive GC skew, green; negative GC skew, violet). The sixth circle represents the GC content (black). The seventh circle represents a Blast against the *E. sibiricum* 255-15 genome. The genome position scaled in 500 kb from base 1 is shown on the inner circle. (**b**) The circular map of plasmid pMCEX (SnapGene, GSL Biotech LLC, Chicago, IL, USA). Orange arrows represent genes coding for proteins with homology in other species from NCBI database, and purple represents hypothetical proteins.

**Figure 2 microorganisms-09-02455-f002:**
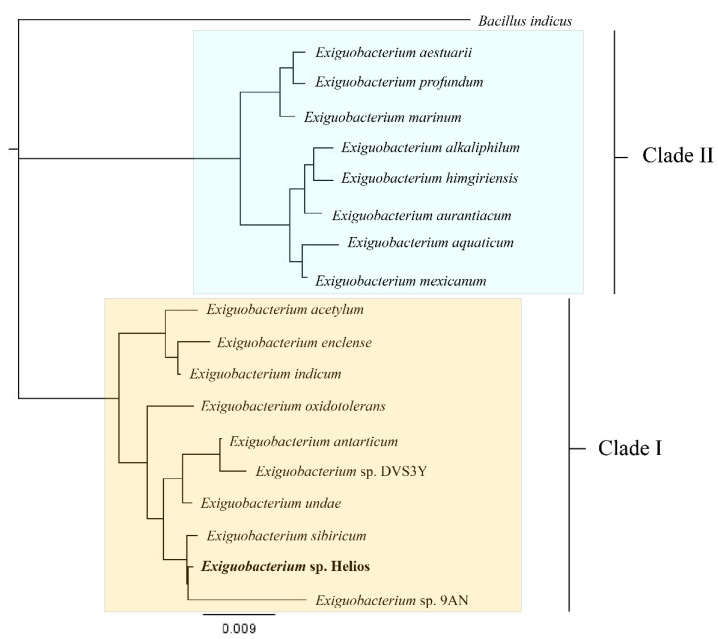
Phylogenetic tree of 16S rRNA genes created by using Geneious v*2020.0*. Accession numbers are as follows: *Exiguobacterium acetylicum* (X70313), *Exiguobacterium aestuarii* (AY594264), *Exiguobacterium alkaliphilum* (EU379016), *Exiguobacterium antarticum* (DQ019164), *Exiguobacterium aquaticum* (JF775503), *Exiguobacterium aurantiacum* (DQ019166), *Exiguobacterium enclense* (JF893462), *Exiguobacterium himgiriensis* (JX999056), *Exiguobacterium indicum* (AJ846291), *Exiguobacterium marinum* (AY594266), Exiguobacterium mexicanum (AM072764), *Exiguobacterium oxidotolerans* (AB105164), *Exiguobacterium profundum* (AY818050), *Exiguobacterium sibiricum* (CP001022), *Exiguobacterium* sp. 9AN (AM072763), *Exiguobacterium* sp. HELIOS (CP053557), *Exiguobacterium* sp. DVS3Y (AY864633), and *Exiguobacterium undae* (DQ019165). Clade I is indicated in pale orange, whereas Clade II is indicated in pale blue.

**Figure 3 microorganisms-09-02455-f003:**
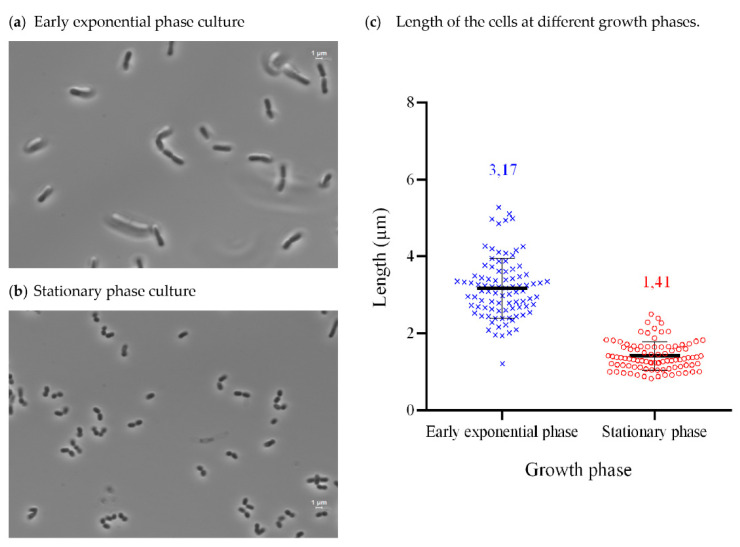
*Exiguobacterium* sp. Helios cells grown in LB at early exponential phase (**a**) and stationary phase (**b**) observed by phase contrast microscopy, 100×. Length of the cells measured with ImageJ in each growth phase (**c**).

**Figure 4 microorganisms-09-02455-f004:**
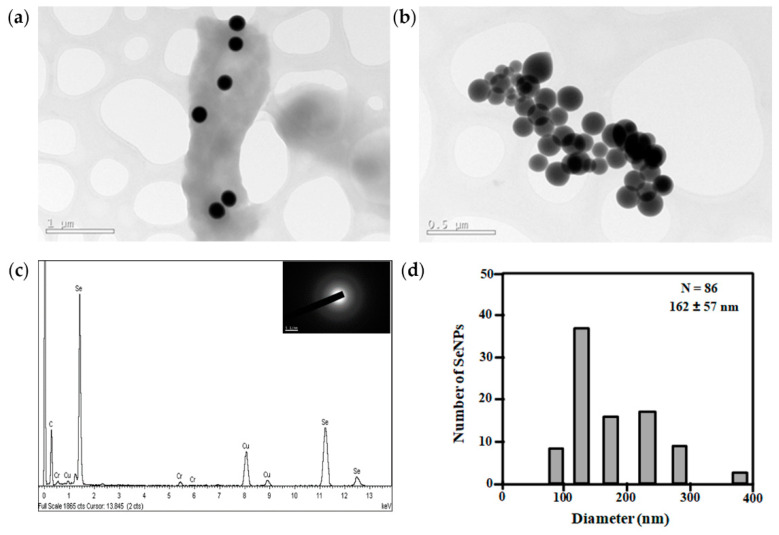
Analysis of the SeNPs production by *Exiguobacterium* sp. Helios cells. (**a**,**b**) TEM analysis showing nanoparticles produced by the bacterial cells. (**c**) EDX analysis of one SeNP from panel **a**. In the inset are shown the diffuse rings in the SAED (selected area electron diffraction) pattern of one SeNP. White line on the inset represents 5 1/nm. (**d**) Size distribution of SeNPs produced by *Exiguobacterium* sp. Helios obtained using ImageJ (N = number of nanoparticles analyzed).

**Figure 5 microorganisms-09-02455-f005:**
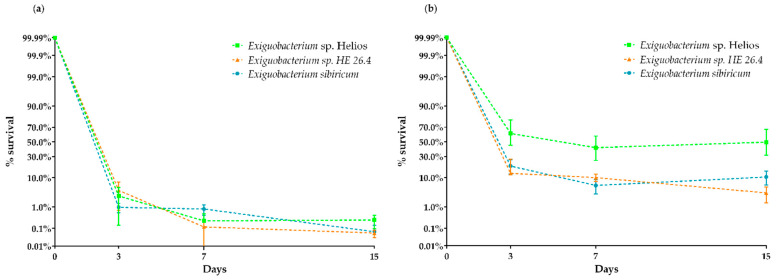
Desiccation test’s survival for *Exiguobacterium* sp. Helios, *E. sibiricum* 255-15*,* and *Exiguobacterium* sp. HE 26.4 at (**a**) early exponential phase and (**b**) stationary phase.

**Figure 6 microorganisms-09-02455-f006:**
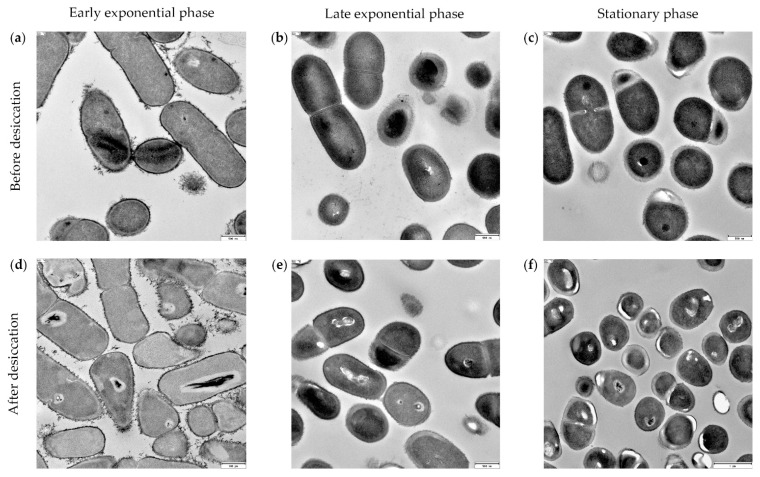
Cells of *Exiguobacterium* sp. Helios observed by TEM. Cells from a culture at early exponential phase before (**a**) and after (**d**) desiccation. Cells from a culture at late exponential phase before (**b**) and after (**e**) desiccation. Cells from a culture at stationary phase before (**c**) and after (**f**) desiccation.

**Table 1 microorganisms-09-02455-t001:** Genes from *Exiguobacterium* sp. Helios putatively involved in competence. The percentage of amino acids identity (%ID aa) is shown.

Gene	Locus	Function Defined in *B. subtilis* subsp. *subtilis* Str. 168	%ID aa	Accession #
*clcC/mecB*	*HNY42_RS01815*	Negative regulator of genetic competence clcC/mecB	80.62%	P37571.1
*comEC*	*HNY42_RS05440*	DNA internalization-related competence protein ComEC/Rec2	28.05%	P39695.2
*comK*	*HNY42_RS05215* *HNY42_RS13920*	Competence transcription factor	24.84% 27.70%	P40396.1
*addABb*	*HNY42_RS05235*	ATP-dependent helicase/nuclease AddAB, subunit B	36.23%	P23477.2
*addABa*	*HNY42_RS05240*	ATP-dependent helicase/nuclease AddAB, subunit A	40.87%	P23478.2
*comEA*	*HNY42_RS05435*	Late competence protein ComEA, DNA receptor	37.44%	P39694.1
*comGA*	*HNY42_RS05970*	Late competence protein ComGA, access of DNA to ComEA	30.68%	P25953.2
*comGB*	*HNY42_RS05975*	Late competence protein ComGB, access of DNA to ComEA	ND	
*comGC*	*HNY42_RS05980*	Late competence protein ComGC, access of DNA to ComEA	ND	
*mecA1*	*HNY42_RS12035*	Adapter protein MecA 1	42.67%	P37958.1
*mecA2*	*HNY42_RS10860*	Adapter protein MecA 2	29.29%	P50734.1
*coiA*	*HNY42_RS12020*	Competence protein CoiA	ND	
*yhgH*	*HNY42_RS14090*	Competence protein F homolog. Protein YhgH required for utilization of DNA as sole source of carbon and energy	32.61%	P39147.1
*comF*	*HNY42_RS14095*	ComF operon protein A, DNA transporter ATPase.	41.42%	P39145.1
*cinA*	*HNY42_RS06685*	competence/damage-inducible protein A	46.68%	P46323.3

**Table 2 microorganisms-09-02455-t002:** Similar sporulation genes found in the genome of *Exiguobacterium* sp. Helios. The percentage of amino acids identity (%ID aa) is shown.

Gene	Locus	Accession #	Function Defined in *B. subtilis* subsp. *subtilis* Str. 168	%ID aa	Accession #
*sigB*	*HNY42_RS00075*	QNR19439.1	RNA polymerase σB factor. General stress protein	63.18%	P06574.3
*spoIIAA*	*HNY42_RS00085*	QNR19441.1	Regulation of the sigma factor (σF) activity.	48.62%	P17903.1
*ykvl*	*HNY42_RS00585*	QNR19540.1	7-carboxy-7-deazaguanine (CDG) synthase. tRNA modification.	49.79%	O31677.1
*yyaC*	*HNY42_RS01240*	QNR22337.1	Spore specific protease.	48.90%	P37521.1
*spoVT/abrB*	*HNY42_RS01560*	QNR19702.1	Transition state regulatory protein AbrB	81.11%	P08874.1
*spoVG*	*HNY42_RS01605*	QNR19711.1	Regulator required for spore cortex synthesis.	65.59%	P28015.1
*mcsA*	*HNY42_RS01805*	QNR19736.1	Activator of protein kinase McsB.	26.70%	P37569.1
*mcsB*	*HNY42_RS01810*	QNR19737.1	Protein arginine kinase.	49.13%	P37570.1
*sigH*	*HNY42_RS01870*	QNR19748.1	RNA polymerase σH. Transcription of early stationary phase genes (sporulation, competence).	71.50%	P17869.1
*dapG*	*HNY42_RS03930*	QNR20129.1	Aspartokinase	40.30%	P08495.2
*yqhQ*	*HNY42_RS06055*	QNR20509.1	Survival to stress conditions. σB and σF regulons.	55.16%	P54515.2
*spo0A*	*HNY42_RS06130*	QNR20524.1	Coordinates DNA replication and initiation of sporulation by binding to sites close to the oriC.	47.47%	P06534.1
*spoVS*	*HNY42_RS06705*	QNR20638.1	Regulator required for dehydration of the spore core and assembly of the coat.	87.21%	P45693.1
*spoIIIAA*	*HNY42_RS12180*	QNR21664.1	Uncharacterized AAA domain-containing protein YrvN	48.12%	O34528.1
*ytvI*	*HNY42_RS13260*	QNR21869.1	Unknown	38.58%	O34991.1
*cwlD*	*HNY42_RS14150*	QNR22035.1	N-acetylmuramoyl-L-alanine amidase, spore cortex peptidoglycan synthesis.	24.19%	O32041.1

**Table 3 microorganisms-09-02455-t003:** Growth of *Exiguobacterium* sp. Helios and *E. sibiricum* 255-15 in different carbon sources. (+) Cells reach an OD_600_ > 0.4. (+/−) Cells reach an OD_600_ > 0.2 and <0.4. (−) Cells reach an OD_600_ < 0.2.

Substrates	*Exiguobacterium* sp. Helios	*E. sibiricum* 255-15
Arabinose	+	−
Fructose	+	+
Galactose	+	+
Glucose	+	+
Lactose	+	+/−
Maltose	+	+
Ribose	+	+
Sucrose	+	+
Xylose	+	−
3,4-Dihydroxybenzoic acid	+	+
3-Hydroxybenzoic acid	−	−
4-Hydroxybenzoic acid	+	+
Benzoic acid	+	+
Gentisic acid	−	−
Phenylacetic acid	+	+
Catechol	−	−
Citric acid	+	+
Pyruvic acid	+	+
Succinic acid	+	+

**Table 4 microorganisms-09-02455-t004:** Antibiotic resistance. MIC concentrations for *Exiguobacterium* sp. Helios.

Antibiotics	MIC (µg/mL)
Kanamycin	6
Gentamycin	10
Ampicillin	50
Chloramphenicol	8
Spectinomycin	75

**Table 5 microorganisms-09-02455-t005:** Metal resistance. MIC concentrations for *Exiguobacterium* sp. Helios.

Metal and Metalloid	MIC (mM)
NiCl_2_	1.25
ZnCl_2_	2.50
K_2_TeO_3_	0.62
K_2_SeO_3_	20.00
NaAsO_2_	1.25
Na_3_AsO_4_	25.00
CdCl_2_	0.62
AgNO_3_	0.62
Pb(NO_3_)_2_	5.00

## Data Availability

The datasets generated in this work can be found in the NCBI under the BioProject PRJNA632476.

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
