# Peer review of "Xerotolerance: A New Property in Exiguobacterium Genus"

_microorganisms, 2021, doi:10.3390/microorganisms9122455_

Round 1

Reviewer 1 Report

In Section 2.4 I have 2 remarks: What was the rationale of choosing the specific primer set instead of either amplifying the entire 16S region or the most commonly used v3-v4 region.
Also, it appears as the authors did Sanger sequencing. In this case, I would suggest using a different term than "amplicon" as it may confuse readers into thinking amplicon metagenomics.
I suppose the authors used blastn against the non-redundant nucleotide collection. Considering the different options and setting that the webtool offers, I would like the authors to be explicit in what they used.

There are several things that are missing and I do not understand from the sequencing and bioinformatics part:
I understand that the authors used a commercial service. It would be nice to briefle mention the tools used in their pipeline. In this case I searched their webpage and found out they are using spades. This is fair. However, and this is very important, basic statistics for the assembly are missing: How many contigs, N50, how many reads map back to the assembly, BUSCO completeness, QUAST results? To make things even more confusing, the authors decided to use a closely related complete genome as scaffold for their isolate. In my experience, this is very dangerous, also taking into account the "manual editing" with no further information. The resulting assembly may be correct but it may also be a forced artifact of the method. Once again, however, metric for the "final assembly" are missing: Coverage, % of reads mapping back to the assembly? Nanopore sequencing (provided as an option by the same sequencing provider) or PacBio (any long read sequencing technology), or multiple pair-end and mate-pair Illumina libraries should be the only way to deduce complete genomes of novel species, unless one is working with mutated strains from the original strain.

The remaining text is well written, but for me, the bioinformatics part is problematic. The cloning tool may work, but the genome may not be what the authors hope it to be.

I would ask for a major revision on at least that part. If the completeness of the genome is not necessary for the cloning approach since plasmids are being used, then perhaps the authors should leave the assembly as draft, or invest in long-read sequencing.

I would be happy to receive a new version of the manuscript, which is indeed interesting. I would also suggest to start the Results section with the comparative genomics, which is the basis of all other analyses (that is the nucleotide sequences themselves).

Author Response

Reviewer 1:

In Section 2.4 I have 2 remarks:

What was the rationale of choosing the specific primer set instead of either amplifying the entire 16S region or the most commonly used v3-v4 region.

When we isolated the strain, we tried to amplify the 16S region with several pairs of primers available in our laboratory that amplify the entire or part of the 16S region, but the only ones that worked at the time were the 28F-519R. Later, we confirmed the genus of the strain with the complete genome sequence and we used the whole 16S region to stablish its phylogenetical position.

Also, it appears as the authors did Sanger sequencing. In this case, I would suggest using a different term than "amplicon" as it may confuse readers into thinking amplicon metagenomics.

The word amplicon was changed as suggested byPCR product."

I suppose the authors used blastn against the non-redundant nucleotide collection. Considering the different options and setting that the webtool offers, I would like the authors to be explicit in what they used.

The following sentence have been added to clarify this point “The resulting sequence was compared to the nucleotide collection at NCBI using the BLAST tool (http://blast.ncbi.nlm.nih.gov/Blast.cgi) optimized for highly similar sequences (megablast)”

There are several things that are missing and I do not understand from the sequencing and bioinformatics part: I understand that the authors used a commercial service. It would be nice to briefly mention the tools used in their pipeline. In this case I searched their webpage and found out they are using spades. This is fair. However, and this is very important, basic statistics for the assembly are missing: How many contigs, N50, how many reads map back to the assembly, BUSCO completeness, QUAST results?

We have included the information requested.

To make things even more confusing, the authors decided to use a closely related complete genome as scaffold for their isolate. In my experience, this is very dangerous, also taking into account the "manual editing" with no further information. The resulting assembly may be correct but it may also be a forced artifact of the method. Once again, however, metric for the "final assembly" are missing: Coverage, % of reads mapping back to the assembly? Nanopore sequencing (provided as an option by the same sequencing provider) or PacBio (any long read sequencing technology), or multiple pair-end and mate-pair Illumina libraries should be the only way to deduce complete genomes of novel species, unless one is working with mutated strains from the original strain.

We apologize for not being clear enough in this regard. We have included the information requested. Moreover, as suggested, we have sequenced the genome using Nanopore technology to confirm the assembly.

The remaining text is well written, but for me, the bioinformatics part is problematic. The cloning tool may work, but the genome may not be what the authors hope it to be. I would ask for a major revision on at least that part. If the completeness of the genome is not necessary for the cloning approach since plasmids are being used, then perhaps the authors should leave the assembly as draft, or invest in long-read sequencing. I would be happy to receive a new version of the manuscript, which is indeed interesting. I would also suggest to start the Results section with the comparative genomics, which is the basis of all other analyses (that is the nucleotide sequences themselves).

Done as suggested.

Reviewer 2 Report

Microorganisms-1418287

In this manuscript, a study on a bacterial strain isolated from a solar panel and characterized by polyestremophiles features was carried out. The identification of the isolate was conducted by 16SrRNA sequencing, assigning it to the genus Exiguobacterium. Features of the isolate were pointed out, including a novelty consisting in xerotolerance, a feature that has not been previously described as a characteristic of the Exiguobacterium genus. The mechanism of xerotolerance was pointed out. Further investigation evidenced the capability to produce nanoparticles when the bacterial strain grew in the presence of selenite added to the culture medium.

The manuscript is well written and offers important information on a new isolate showing polyextremophile features, including the capability to produce Se-nanoparticles as a mechanism of resistance to this metalloid. The isolated strain includes a novelty, in evidencing for the first time xerotolerance in a strain beloging to this genus. The possible and convenient exploitation of the isolate in biotechnological purposes is proposed.

Revisions

Line 27: ‘nanoparticules’ change to nanoparticles;

line 38: ‘Bacillae’ change to ‘Bacillales’;

line 203: ‘volume in microtiters;

line 295: Table 3, as Se-resistance and As-resistance are related, maybe the same mechanisms of resistance is present for both metalloids;

Table 4 and Table S2: genes must be reported in Italic style;

Table 5S: the names of Families need to be reported in Italic style.

Author Response

Reviewer 2:

Line 27: ‘nanoparticules’ change to nanoparticles; Corrected

line 38: ‘Bacillae’ change to ‘Bacillales’; Corrected

line 203: ‘volume in microtiters; Corrected

line 295: Table 3, as Se-resistance and As-resistance are related, maybe the same mechanisms of resistance is present for both metalloids;

The analysis of the genome provided several genes related with arsenate/arsenite resistance (i.e., one glutaredoxin-dependent arsenate reductase (ArsC); two arsenite transporters (ArsB and Acr3) and three ArsR-like transcripcional regulators). We did not find any gene for the biosynthesis of arsenomethylated or arsenoorganic compounds. The molecular mechanisms for selenite resistance described so far might depend of each strain. In this sense, it has been proposed that general reductases such as for instance nitrite reductase NirK, fumarate reductases or others can be involved in the reduction of selenite to elemental selenium. In any case, to the best of our knowledge, the mechanisms underlying the bacterial resistance to arsenic and selenite appears to be different in all cases that have been described in the literature so far.

Table 4 and Table S2: genes must be reported in Italic style; Corrected

Table 5S: the names of Families need to be reported in Italic style. Corrected

Round 2

Reviewer 1 Report

The authors have done a satisfactory effort to address all concerns. The manuscript appears complete and I have no further comments.

Author Response

Thank you for your reviewing.